# Insights into Hyperparathyroidism–Jaw Tumour Syndrome: From Endocrine Acumen to the Spectrum of *CDC73* Gene and Parafibromin-Deficient Tumours

**DOI:** 10.3390/ijms25042301

**Published:** 2024-02-15

**Authors:** Ana-Maria Gheorghe, Oana-Claudia Sima, Alexandru Florin Florescu, Adrian Ciuche, Claudiu Nistor, Florica Sandru, Mara Carsote

**Affiliations:** 1PhD Doctoral School of “Carol Davila”, University of Medicine and Pharmacy, 050474 Bucharest, Romania; anamaria.gheorghe96@yahoo.com (A.-M.G.); oanaclaudia1@yahoo.com (O.-C.S.); 2Endocrinology Department, “Grigore T. Popa” University of Medicine and Pharmacy, 700111 Iasi, Romania; alexandru-florin.florescu@umfiasi.ro; 3Endocrinology Department, “Sf. Spiridon” Emergency County Clinical Hospital, 700111 Iasi, Romania; 4Department 4—Cardio-Thoracic Pathology, Thoracic Surgery II Discipline, Faculty of Medicine, “Carol Davila” University of Medicine and Pharmacy, 050474 Bucharest, Romania; adrianciuche@gmail.com; 5Thoracic Surgery Department, “Dr. Carol Davila” Central Emergency University Military Hospital, 010825 Bucharest, Romania; 6Department of Dermatovenerology, “Carol Davila” University of Medicine and Pharmacy, 050474 Bucharest, Romania; 7Department of Dermatovenerology, “Elias” University Emergency Hospital, 011461 Bucharest, Romania; 8Department of Endocrinology, Faculty of Medicine, “Carol Davila” University of Medicine and Pharmacy, 050474 Bucharest, Romania; carsote_m@hotmail.com; 9Department of Clinical Endocrinology V, “C.I. Parhon” National Institute of Endocrinology, 020021 Bucharest, Romania

**Keywords:** *CDC73*, hyperparathyroidism–jaw tumour syndrome, gene, parathyroid tumour, parathyroid carcinoma, jaw tumour, parathyroidectomy, parafibromin, immunohistochemistry

## Abstract

A total of 1 out of 10 patients with primary hyperparathyroidism (PHP) presents an underlying genetic form, such as multiple endocrine neoplasia types 1, 2A, etc., as well as hyperparathyroidism–jaw tumour syndrome (HJT). We aimed to summarise the recent data, thus raising more awareness regarding HJT, from the clinical perspective of PHP in association with the challenges and pitfalls of *CDC73* genetic testing and parafibromin staining. This narrative review included a sample-focused analysis from the past decade according to a PubMed search. We identified 17 original human studies (≥4 patients per article). The mean age at disease onset was between 20.8 and 39.5 years, while the largest study found that 71% of patients had HJT recognised before the age of 30. Males and females seemed to be equally affected, in contrast with sporadic PHP. PHP represented the central manifestation of HJT, occurring as the first manifestation in up to 85% of HJT cases. A biochemistry panel found a mean serum calcium level above the level of 12 mg/dL in PHP. PTH was elevated in HJT as well, with average values of at least 236.6 pg/mL. The most frequent pathological type in PHP was a parathyroid adenoma, but the incidence of a parathyroid carcinoma was much higher than in non-HJT cases (15% of all parathyroid tumours), with the diagnosis being established between the age of 15 and 37.5. In some families up to 85% of carriers suffered from a parathyroid carcinoma thus indicating that certain *CDC73* pathogenic variants may harbour a higher risk. An important issue in HJT was represented by the parafibromin profile in the parathyroid tumours since in HJT both parathyroid adenomas and carcinomas might display a deficient immunoreactivity. Another frequent manifestation in HJT was ossifying fibromas of the jaw (affecting 5.4% to 50% of patients; the largest study found a prevalence of 15.4%). HJT was associated with a wide variety of kidney lesion (mostly: kidney cysts, with a prevalence of up to 75%, and renal tumours involved in 19% of patients). The risk of uterine lesions seemed increased in HJT, especially with concern to leiomyomas, adenofibromas, and adenomyosis. The underlying pathogenic mechanisms and the involvement of *CDC73* pathogenic variants and parafibromin expression are yet to be explored. Currently, the heterogeneous expression of parafibromin status and, the wide spectrum of *CDC73* mutations including the variety of clinical presentations in HJT, make it difficult to predict the phenotype based on the genotype. The central role of HJT-PHP is, however, the main clinical element, while the elevated risk of parathyroid carcinoma requires a special awareness.

## 1. Introduction

Primary hyperparathyroidism (PHP), affecting 1–2% of individuals over 55 years of age, with females being more prone [1,2], encompasses a varied panel of presentations, from those easily treatable using a modern facelift approach, namely completely asymptomatic cases and even normocalcemic patterns (with incidental detection amid different biochemistry and endocrine screening protocols), to the challenging spectrum of familial/genetic PHP. This may display not only a more severe clinical manifestation, a higher risk of multi-glandular disease, and an onset at younger ages when compare to sporadic PHP but also the burden of distinct non-parathyroid comorbidities that lower the overall quality of life via endocrine and non-endocrine neoplasia [3,4,5]. 

Approximately 1 out of 10 patients with PHP presents a genetic form (also called monogenetic or hereditary PHP), underlying multiple endocrine neoplasia (MEN) types 1, 2A, and 4 and non-syndromic (familial) PHP, including neonatal severe PHP and familial hypocalciuric hypercalcemia, as well as hyperparathyroidism–jaw tumour syndrome (HJT) [6,7,8].

HJT represents a complex disorder, not being only a familial type of PHP but also a parafibromin-deficient parathyroid neoplasm, a neoplasia recently recognised as a distinct entity by the World Health Organization (WHO) [9]. HJT is caused by a germline pathogenic variant of the *CDC73* (cell division cycle 73) gene, with an autosomal-dominant inheritance and variable penetrance [10]. Of note, the connection between the *CDC73* gene (named *HRPT2* at the time) and HJT was established two decades ago [11], and a heterogeneous frame has been reported so far, with the abundance of data being still an open issue. The *CDC73* gene encodes parafibromin, a protein that acts as a tumour suppressor; parafibromin interacts with different proteins through its C-terminal domain, being a part of the polymerase II-associated factor 1 (PAF1) complex that binds to RNA polymerase II and blocks cyclin D1 expression [11,12,13,14]. A loss of nuclear parafibromin staining in HJT-associated parathyroid tumours is caused by pathogenic *CDC73* variants [15]. 

The most common manifestation of HJT is PHP; in the majority this is a single-gland disease of the adenoma type (despite being a genetic form of PHP). However, parathyroid carcinoma has been described as well, and although it is a rare disease in the general population [16], it occurs at much higher rates in patients with HJT than non-HJT individuals [17,18]. While no multidisciplinary guideline specifically addresses the issue of HJT, the current recommendations for HJT-related PHP include an early parathyroidectomy due to the elevated risk of a parathyroid carcinoma under these specific circumstances [19,20]. Considering that prompt recognition of genetic PHP is the key to timely diagnosis of high-risk tumours such as HJT-associated parathyroid carcinomas, molecular and immunohistochemistry markers are essential and serve as prognostic markers amid our modern era of combined endocrine–genetic–surgery approaches [20,21]. 

Apart from the parathyroid neoplasms (adenomas, atypical adenomas, or carcinomas), HJT includes jaw tumours and kidney lesions, such as Wilms tumours and kidney cysts, as well as uterine conditions [22]. Jaw tumours occur in 25–50% of HJT cases being histologically distinct from bone brown tumours or parathyroid carcinoma-derived metastases. They consist of ossifying fibromas, affect both the mandible and the maxilla (with a predilection for the mandible) [23,24,25], and may sometimes reach impressive sizes and compress nearby structures [26]. Their management involves a complete surgical resection, followed by reconstruction in some cases [27]. 

The lack of syndromic manifestations other than PHP in one individual or one family qualifies PHP as a familial isolated PHP associated with *CDC73* pathogenic variants since there is no clear genotype–phenotype correlation in patients with germline *CDC73* pathogenic variants. One argument is the association of the same mutation with either HJT, familial isolated PHP, or sporadic parathyroid carcinoma in different patients, etc. [13,28]. Other authors, however, consider that *CDC73*-associated familial PHP is actually an incomplete form of the syndrome [13,29]. Larger genetic studies are mandatory for a better understanding of the *CDC73*-connected clinical and molecular spectrum, including parafibromin’s role, as reflected by the expression “para this, fibromin that” (thus showing its complexity and unexpected implications and interferences), specifically with regard to the HJT profile and associated genetic picture [13].

The current work aimed to summarise the recent data on HJT, thus bringing more awareness regarding this condition, especially from the clinical perspective of PHP in association with the challenges and pitfalls of *CDC73* genetic testing and parafibromin staining. This was a narrative review underlying a sample-focused analysis of the last decade; particularly, we conducted a PubMed-based search of English language medical literature from January 2014 until October 2023 regarding HJT. We used search keywords (in different combinations) such as “hyperparathyroidism jaw tumour syndrome” (alternatively, “hyperparathyroidism” or “parathyroid”) and “jaw neoplasm”, “jaw syndrome”, “parafibromin”, or “*CDC73*”. We included full-length original studies (and case series) studying at least four individuals per paper that provided *CDC73* genetic testing and excluded case reports (<four patients/case), experimental, and non-human studies. 

## 2. Sample-Focused Analysis

We identified 17 original studies regarding the spectrum of *CDC73*-associated HJT or HJT-PHP, having as a starting point either the HJT population or subjects with PHP who underwent genetic assays or screening within one family [30,31,32,33,34,35,36,37,38,39,40,41,42,43,44,45,46] (Table 1).

### 2.1. HJT Syndrome: Patients’ Characteristics at Presentation

Most of the enrolled patients were diagnosed before the age of 40 years, with the youngest subject being 10 [31] and the oldest 71 [35]. While some studies found that males were more affected than females [31,34,46], others identified quite the opposite results [35,41] or a similar sex ratio [30,32,43]. Overall, no specific sex distribution was confirmed (of note, the type of studied population varied among individuals, such as initial admission for HJT/PHP or one family screening). For example, Tora et al. [31] studied a retrospective cohort (N = 68) with an age at first admission between 10 and 30 years, and 71% (N = 36) of them being women [31]. Iacobone et al. [35] identified an older age at diagnosis of HJT-related PHP (a median of 38, ranging between 11 and 71 years; female-to-male ratio of 14/8) [35]. Figueiredo et al. [30] showed an average age of 33.3 ± 13.1 years at HJT diagnosis and a mean of 35.1 ± 13.6 years with respect to PHP onset, also identifying an equal distribution among women and men [30]. Similarly, Mehta et al. [46] showed a mean age at HJT diagnosis of 30.7 years (between 18 and 49), with 62.5% (N = 10) of them being males [46]. 

Le Collen et al. [34] reported data from the largest kindred study, comprising 24 genetically tested members out of 47. A total of 54.16% (N = 13) of these tested members were positive for a *CDC73* large deletion, and 61.5% of all carriers (N = 8) were symptomatic in terms of either PHP and/or ossifying fibromas of the jaw. Their median age at PHP onset was 37 ± 13.6 years [34]. Another age derivate analysis from 2017 was provided by an observational cohort of 65 patients with PHP: at the moment of HJT diagnosis, the subjects were aged between 13 and 24 years (with an average of 20.8 years). It should be noted, however, that an age under 40 was an inclusion criterion in this specific study. Most of the patients were females (83.3%, N = 5), and there was only one male [41]. Moreover, Khadilkar et al. [43] analysed seven cases out of four families with familial PHP associated with *CDC73* variants, and three out of the four families were confirmed to have HJT, while the fourth family (N = 3 family members) was diagnosed with isolated PHP; across this sample of seven individuals, four subjects (sex ratio of 1) were found to be HJT-positive, with an average age of 36 years [43]. Also, Yang et al. [32] studied a family with PHP caused by a *CDC73* variant and found out that 80% (N = 4) of the members with *CDC73* pathogenic variants developed PHP at an average age of 39.5 years; the sex ratio of these individuals was 1 [32].

### 2.2. HJT-Associated PHP

PHP represented the most common clinical presentation regardless of the endocrine or non-endocrine pattern amid HJT; the majority of the studies agreed that it usually represented the first clinical element of the syndrome [30,31,35]. Although most patients were asymptomatic and diagnosed according to active surveillance or routine testing [30], some of them had clear symptoms based on the main clinical PHP clusters as generally described [47]: neurologic, digestive, renal or even osseous-like fragility fractures [46]. Particularly, we mention again the (rather large) study (that was published in 2023) on 68 subjects with HJT pertaining to 29 kindred; a total of 85% of the affected subjects had PHP as the initial clinical anomaly of HJT at a median age of 26. PHP detection either occurred due to either an active surveillance protocol (noting a positive family history of hypercalcemia) in one third of the patients or due to routine blood assays (in 22% of them), while the remaining 35% were confirmed starting from specific complains in relationship with symptomatic PHP (which, overall, was caused by a parathyroid adenoma in 65%, respectively, and carcinoma in 35% of cases) [31]. Another cohort of 16 people (belonging to seven families with HJT) showed that all of them were symptomatic, mostly (43.8%) with regard to neurologic features such as headaches, memory loss, and difficulty concentrating, followed by fatigue (37.5%), gastrointestinal symptoms like abdominal pain, constipation, gastroesophageal reflux (31.3%), polyuria, and/or nephrolithiasis (18.8%); bone -related pain (18.8%), and osteoporotic fractures (12.5%) [46]. 

The largest family screening-based assessment (in terms of 47 members, who were evaluated across one observational study published in 2021) showed that 13/47 of them were *CDC73* carriers, and 7/13 were confirmed to have PHP (male-to-female ratio of 5 to 2). The remaining individuals (6/13) had only a jaw tumour (N = 2) or were healthy carriers (N = 4). The youngest subject with HJT-related PHP was 13 and the oldest was 54 (a median of 39 was confirmed). The PHP-associated clinical panel pinpointed renal complications in 57.1% of them (4/7), namely urinary lithiasis, nephrocalcinosis, and chronic kidney failure. Of note, no case of parathyroid malignancy was described in this mentioned family as an originating source of PHP [34]. A retrospective analysis from 2020 on 37 patients with HJT showed a PHP prevalence of 59.5% (22/37) having the highest incidence in individuals older than 30 years (68%, 15/22) [35], while another similar cohort study on 20 people with HJT revealed that 17/20 of them, representing 85%, had PHP and 23.5% of these cases had a parathyroid carcinoma [30]. 

Another case series published in 2019 showed that half of the screened family members (3/6) were diagnosed with PHP (two males and one female) at the ages of 24, 35, and 46, respectively; notably 2/3 of these cases had an associated malignant pathologic report of the parathyroid gland, while the other subject had a double parathyroid adenoma. All three patients with HJT-PHP carried the pathogenic variant c.128-IVS1 + 1 delG of the *CDC73* gene. Additionally, the female subject with parathyroid carcinoma-associated PHP had a jaw ossifying fibroma and a uterine fibroid [38]. 

#### 2.2.1. Lab Findings in PHP

A few data among this 10-year sample-based study provided a pre-operatory biochemistry and hormonal panel for HJT-PHP. For example, a lab profile was provided in 20 patients diagnosed with PHP according to one study from 2020: the serum total calcium levels were between 10.26 mg/dL (2.56 mmol/L) and 18.31 mg/dL (4.57 mmol/L), with a mean of 12.9 mg/dL (3.24 mmol/L), and the mean PTH (parathyroid hormone) levels were increased by a mean 2.29-fold [35]. Another cohort from 2014 (N = 16 subjects with HJT-PHP) pointed out an average total calcium of 12.2 mg/dL (median of 12.0 mg/dL, normal levels between 8.5 and 10.2 mg/dL) and an average PTH of 236.6 pg/mL (median of 190.4 pg/mL, normal values between 10 and 55 pg/mL) [46]. 

An observational study from 2021 (N = 7 patients with PHP) showed that baseline calcaemic values at PHP diagnosis ranged between 10.82 mg/dL (2.7 mmol/L) and 17.95 mg/dL (4.48 mmol/L), with a mean of 12.54 ± 2.8 mg/dL (3.13 ± 0.7 mmol/L) in association with PTH levels between 78 pg/mL and 1100 pg/mL [34], while another one with a similar study design identified calcium levels between 11.78 mg/dL (2.94 mmol/L) and 18 mg/dL (4.49 mmol/L) at first presentation, with an average of 15 mg/dL (3.74 mmol/L) and PTH between 125.1 pg/mL and 2440 pg/mL (average of 1324.65 pg/mL) these data being the highest values found in pre-operatory assays among all mentioned cohorts in HJT-PHP (a 200-fold average PTH increase versus normal) [41]. Khadilkar et al. [43] also confirmed very elevated mean PTH of 838.75 pg/mL [43]; Yang et al. [32] conducted a study on a family (N = 5 individuals who were *CDC73* variant carriers) and 80% (N = 4) of them had PHP with an average calcium of 10.62 mg/dL (2.65 mmol/L) and PTH of 411.25 pg/mL (Table 2) [32].

#### 2.2.2. PHP Management: From Preoperative Localization to Parathyroidectomy

Two studies addressed the issue of localization before surgery with concern to the parathyroid tumours [35,46]. One of them compared preoperative imaging results provided by using the neck ultrasonography and/or Sestamibi scan and confirmed that single-gland and multi-glandular disease were correctly identified (as confirmed by the post-operatory histological report) in 92.8% of cases (single and multiple gland involvement was found in 11 and 2 subjects, respectively); Sestamibi scan associated 46.2% accuracy in pre-surgery prediction rate of lateralization [46]. Another analysis showed that 70% of one cohort with HJT-PHP had concordant pre-surgery localization (data on the specific imaging techniques were not available), and 25% of the patients had discordant results, while 5% (representing a single case) had a negative result; notably, all three patients, whose parathyroid carcinoma diagnoses were confirmed post-operatively, were within the first mentioned subgroup [35]. Interestingly, discordant localization outcome correlated with an increased post-parathyroidectomy recurrence rate [35], as generally known in non-HJT cases of PHP [48]. 

Most HJT-PHP individuals underwent surgery; bilateral neck exploration was frequently chosen as start-up approach [35,46]; unilateral parathyroidectomy was preferred as the disease mostly was of single-gland type [30,35,43,46]. Reoperations were performed either due to the recurrence of parathyroid carcinoma [35] or due to asynchronous adenomas [35,43]. Among post-parathyroidectomy complications we noted transient or permanent hypoparathyroidism [30,35,46], and recurrent nerve palsy [35,46]. Particularly, we mention that among the 20 patients with HJT-PHP, 65% of them underwent bilateral neck explorations, 35% had targeted unilateral neck exploration, while the single case with multi-gland disease followed a procedure of subtotal parathyroidectomy. Cure rate was of 100% for benign tumours (17/20), except the fact that for, during follow-up, a recurrence in one subject who experienced a second single tumour (1/17) required re-do surgery; upon carcinoma resection, persistent or recurrent PHP was found in 66% (N = 2) of patients. Surgical complications included recurrent nerve palsy (N = 2) and transient hypoparathyroidism (N = 5), especially due to bilateral neck exploration [35]. 

Figueiredo et al. [30] showed that 58.8% of the HJT-PHP subjects underwent selective parathyroid resection, 17.6% had a subtotal parathyroidectomy, while a total parathyroidectomy was performed in 11.8% (including a case with self-transplantation) of them. Persistent and recurrent post-operatory hyperparathyroidism rates were of 12.5%, and 25%, respectively; hypoparathyroidism was identified in 31.3% of cases [30]. Another cohort (N = 55 patients with HJT-PHP) revealed that more than half underwent parathyroid surgery and most of them (83.33%) suffered a re-intervention [31]. The results from a cohort (N = 16) with HJT-PHP were the followings: all had surgery, 93.75% of them underwent bilateral neck exploration as initial surgical approach; subtotal parathyroidectomy was performed in 37.5% of patients; post-operatory hypoparathyroidism, either transient (62.5%) or permanent hypoparathyroidism (12.5%) was reported, as well [46]. While 83.3% (5/6) of one case series achieved control of PHP following surgery, one patient had persistent disease due to multiple lung metastases [41]. Another case series of seven family members with HJT-PHP showed that all were referred to surgery and had a single gland disease (histological reports confirmed: one adenoma, two atypical adenomas and four cystic adenomas) [34].

#### 2.2.3. Parathyroid Carcinoma

Parathyroid carcinoma had a high prevalence in most studies compared to the expected rate from the general population, the highest rate being of 83.3% (N = 5/6 patients) [41]. Other prevalence results were of 31% of HJT-PHP cases [31] and 37.5% (N = 6/16), respectively, with metastatic disease in two thirds of these persons and outcome toward death due to complications in 50% of patients (N = 3/6). The overall median survival of the subjects with parathyroid carcinoma amid HJT-PHP was of 8.9 years. Compared to non-malignant parathyroid tumours, pre-operatory clues included higher serum calcium versus adenomas [46]. Lower rates of HJT-PHP-related parathyroid carcinoma were observed according to the following 4 studies: 25% (meaning a single case that remained uncured following surgery) [43]; 23.5% (N = 4) [30]; 20% of the carriers (N = 1) belonging to one family with PHP caused by a *CDC73* pathogenic variant [32]; and 15% (N = 3) [35] representing the lowest rate of parathyroid malignancy across all studies. In this instance, one patient died due to metabolic complications of hypercalcemia caused by persistent PHP 2.5 years following surgery, while another had persistent metastatic disease five years post-surgery, and one individual was cured and had a 2-year disease free interval following the surgical procedure [35].

#### 2.2.4. Parafibromin Staining

According to the mentioned methods, we identified 7/17 studies that explored the immunohistochemistry expression of nuclear parafibromin in parathyroid tumours belonging to the spectrum of HJT-PHP with rather contrasting results, noting an overall low level of statistical significance among these studies to specifically assess this protein status. The largest study was performed by Gill et al. [37] on 815 parathyroid tumours from 789 subjects; 5.2% of the tumours (N = 43) from 5.1% of patients (N = 40) displayed a loss of parafibromin staining. Compared to a control group, these patients had a younger age at disease onset. 66.7% (16/24) of genetically tested patients had pathogenic germline variants of *CDC73* gene (N = 14) or a whole gene deletion (N = 2). However, the inability to rule out variants in the rest of the subjects was a potential bias of this study impeding the establishment of the exact prevalence. Most patients with parafibromin deficient tumours, meaning 66.7%, had adenomas (N = 24). The prevalence of carcinoma among patients with negative parafibromin staining was of 33.3% (N = 16). Among patients with germline variants of the *CDC73* and loss of parafibromin staining, the prevalence of carcinoma was of 25% (N = 4) versus adenomas (75%; N = 12). *CDC73* variants associated with carcinoma and loss of parafibromin staining were c.685_688delAGAGp.(Arg229Tyr)fs*27, IVS7 + 2T > G, c.271C > T, p.(Arg91*) and a whole gene deletion [37]. 

Conversely, Tora et al. [30] did not find a correlation between parafibromin stain and genotype or tumour histology [30]. Mehta et al. [46] found diffuse loss of parafibromin staining in 77.8% (N = 7/9) in the adenomatous component as opposed to the presence of parafibromin staining in the normal surrounding tissue. The remaining patients (32.2%, N = 2) had a preservation of nuclear parafibromin expression. All patients for whom parafibromin staining was analysed had parathyroid adenomas. There was no data available regarding carcinoma cases [46]. 

Other heterogeneous results were a series of three parathyroid tumours showed a weakly positive parafibromin staining in one adenoma, and two negative parathyroid carcinomas [38]; irregular nuclear parafibromin reactivity was confirmed on one parathyroid malignancy, mutant parafibromin pattern being associated with c.1379delT pathogenic variant and 65% loss of its expression along with part of its tumour suppressor effect [44]; loss of parafibromin staining was observed by Le Collen et al. [34] in all five samples belonging to a family with parathyroid adenomas [34]. 

### 2.3. HJT-Related Jaw Tumours 

Patients with HJT were affected by jaw tumours in 5.4% up to 50% of the patients; the studies with larger sample sizes found a prevalence between 5.4 and 15.4% [31,34,35,46], while the data on family members protocols reached higher rates [43,44]. Specifically, the lowest rate was of 5.4% (N = 2/37) with a confirmation of jaw ossifying fibromas [35], while other studies had a higher prevalence of 12.5% (N = 2) [46]; 15.4% of the carriers meaning two patients, the jaw tumour being the sole manifestation in one of the cases [34]; 25% (N = 1) [44], and 50% (N = 2) [43], respectively.

In certain cases, these tumours were the first manifestation of the syndrome [30,31], and exceptionally they were the only clinical elements of the disease [34]. Of note, the jaw involvement as an initial clinical clue represented a point of analysis in these two cohorts: 10% (N = 2) of all subjects [30], respectively, 5% (with an overall median age at presentation of 24 years). Four out of seven jaw masses (57%) were surgically removed. Both tumours (2/4) that had a pathologic report were confirmed as being ossifying fibromas [31].

### 2.4. Other Tumours in HJT

In addition to the mentioned parathyroid and jaw tumours, some other types are worth mentioning, particularly, regarding renal and uterine sites. Of note, a case series of six individuals never identified non-parathyroid, non-jaw tumours [41]. Uterine tumours such as fibroids, adenomyosis, adenomyoma, adenocarcinoma, and polyps affected 38% of the females, while kidney tumours (mixed epithelial stromal tumours and Wilms tumours) were reported in 6% of the patients in one cited study. Most kidney neoplasia (three out of four) occurred in females with concurrent uterine masses. Uterine lesions were the initial manifestation in 18% of the affected females, while kidney tumours were the first clinical element in one single case [31]. Another cohort identified uterine tumours in one third of the women (N = 2/6), respectively, 18.8% of patients (N = 3/16) had renal tumours [46]. The highest rate of uterine involvement reached 60.8% [35].

The spectrum of these lesions in HJT included a renal metastatic tumour (a single case, representing 2.7% of one cohort [35]); kidney cysts in 15.4%, meaning two patients (while one of them had a synchronous uterine polyp) [34], respectively, in 75% of another cohort, meaning three other cases [43]; and one report of a thyroid differentiated micro-carcinoma and prostate malignancy [34]. A distinct presentation was found by Vocke et al. [39] in three members of the same family who associated a germline variant (c.3G > T variant of *CDC73* gene, thus impeding parafibromin translation). All these subjects had mixed epithelial and stromal tumours of the kidney, and all females (N = 2) had uterine fibroids. One of the women (1/2) had bilateral kidney tumours in association with somatic loss of heterozygosity in both lesions [39]. Moreover, a retrospective cohort study by van der Tuin et al. [42] reported various tumours in HJT subjects and *CDC73* variants out of three families. Apart from the parathyroid adenomas (N = 16), and jaw tumours (N = 5), the patients developed renal cysts (N = 5), congenital urinary tract abnormality, Wilms tumour, pancreatic ductal adenocarcinoma (N = 2), Hürthle cell adenoma of the thyroid (N = 1), and mixed germ cell testicular tumour (N = 1) (Table 3) [42].

### 2.5. The Genetic Spectrum in HJT

The gene-related data on HJT yielded very interesting results, and we highlight the key findings. Phenotype–genotype associations were provided by Tora et al. [31]. The patient with a Wilms tumour had biallelic loss comprising a germline *CDC73* variant p.L95P and a pathogenic variant in the trans-allele, p.M1V (c.1A > G). The patients with mixed epithelial stromal tumours had a germline c.3G > T (p.M1I) variant. The germline *CDC73* variant c.164 A > C (p.Y55S) was also associated with kidney tumours. A genotype was not always associated with the same phenotype, due to variable penetrance, generating a wide spectrum of manifestations, from a clinically silent phenotype to a parathyroid carcinoma, in the same kindred, an aspect which represents a major challenge in these patients despite meticulous testing in one family and long-term surveillance [31]. 

Li et al. [36] conducted the largest study that analysed the correlations between genotype and phenotype in patients with parathyroid tumours and *CDC73* pathogenic variants. The study included two cohorts, one original cohort of 68 subjects and a validation cohort from reported cases. In the original cohort, 61.8% (N = 42 subjects) had a confirmed pathogenic variant, while the remaining subjects were presumed positive for pathogenic variants. The prevalence of parathyroid carcinoma was of 20.6% (N = 14). 61.8% (N = 42) of the patients had parathyroid adenomas, while 17.6% (N = 12 of them did not have any parathyroid disease. This study found that high-impact (gross indels, splicing, frameshift, and nonsense) germline variants and variants affecting the C-terminal domain of the parafibromin protein are associated with an increased risk of parathyroid carcinoma and jaw tumours, by disrupting the expression and tumour suppressing activity of parafibromin. High-impact *CDC73* variants were also associated with a larger tumour size and higher levels of PTH [36].

The penetrance of *CDC73* variants among PHP patients and the clinical manifestations of *CDC73*-related diseases, including HJT, were analysed by van der Tuin et al. [42] according to a nationwide Dutch cohort (89 subjects with PHP were referred for genetic testing of *CDC73*). 12.4% (N = 11) were tested positive for pathogenic germline *CDC73* variants. Out of 18 subjects with (suspected) HJT, 17% (N = 3) had a pathogenic germline *CDC73* variants [c.687_688dellAG, p.(Arg229Serfs*37) (N = 1), c.3_15dup, p.(Ser6Glyfs*5) (N = 1), and c.760C.T, p.(Gln254*) (N = 1)]. Relatives of the index cases were also tested. The c.687_688dellAG, p.(Arg229Serfs*37) variant was found in 65% (N = 24) of these relatives. In another family with the c.3_15dup, p.(Ser6Glyfs*5) variant, all tested relatives (N = 3) carried this variant. The index case had both a parathyroid adenoma and jaw tumour, while the relatives had a jaw tumour, as well. Congenital urinary tract abnormality and a Wilms tumour were also identified. Concerning the third mentioned case, diagnosed with a parathyroid adenoma, 50% of the relatives (N = 4) carried the c.760C.T, p.(Gln254*) *CDC73* variant. Only one of them was symptomatic and had a history of a jaw tumour. As bias, lack of referral criteria for genetic testing in PHP group impaired the accuracy of the prevalence data [42].

In a retrospective study by Iacobone et al. [35] on five families with HJT, 62.7% (N = 37/59) of the subjects who underwent clinical and genetic investigation were confirmed with the syndrome. In the genetically tested group, 67.7% were positive; a total of 58% of them were diagnosed with PHP, but 32.2% showed no manifestations of the disease. The condition manifested clinically in 87.5% of subjects who were positive for the frameshift variant c.433_442delinsAGA, and 85.7% of these had PHP. Regarding the missense c.188T > C transition, 57.14% of the positive persons had manifestations of the syndrome, and all of them had PHP. The same frequency of disease manifestations was found in the kindred with the c.136_144 del5 variant. The c.276delA p.Asp93Ilefs*16 variant caused clinical manifestations in 66.7% of them, and half had PHP [35]. Mehta et al. [46] tested 16 patients from seven families with HJT; all of them had variants of the *CDC73* gene, as it was one of the inclusion criteria in this study. One family (N = 7) had a whole-gene deletion, three families (N = 5) had c.687_688dupAG variant, one family (N = 2) had substitution of a serine for tyrosine (p.Tyr55Ser) in exon 2, one patient had a c.664C > T, p.Arg222X variant, and another patient had a c.226C > T, p.Arg76X variant [46]. 

Other variants were confirmed, such as c.271C > T (p.Arg91), c.496C > T (p.Gln166), c.685A > T (p.Arg229), and gross deletion of the whole *CDC73* gene, to be associated with a parathyroid carcinoma [41]. Another 20 HJT cases in seven families displayed four different pathogenic (N = 6) and likely pathogenic (N = 13) variants and a whole-gene deletion (N = 1) of the mentioned gene. The most prevalent variant was c.356del p.(Gln119Argfs*14) [31]. Additionally, four different variants in HJT-PHP were identified. One family was diagnosed with isolated PHP due to lack of other syndromic manifestations and associated a codon 222CGA(Arg) > TGA variant [43]. In the largest family harbouring the same likely pathogenic variant reported, 13 out of 24 members that were tested carried a large deletion, c.(237 + 1_238-1)_(307 + 1_308-1)del;p. [34]. Another *CDC73* variant reported in the study conducted by De Luise et al. [33] was a large germline deletion of the first 10 exons in a family whose members developed numerous oncocytic parathyroid tumours: half of them had carcinomas (N = 2) and the other half had adenomas. The subjects were also affected by a pathogenic germline variant, m.2356A > G of the mitochondrial DNA, as well as different somatic mitochondrial somatic pathogenic variants (m.14973G > A, m.5147G > A, m.3380G > A, m.14387A > G, m.10371G > A) [33]. Of note, c.3G > T variant of the *CDC73* gene was associated with a particular presentation: mixed epithelial and stromal tumours of the kidney in all carriers (N = 3), and uterine fibroids in all female carriers (N = 2) [39].

## 3. Discussion

### 3.1. Clinical Issues and Panel of Investigations in Studies concerning HJT

The sample—focused analysis included 17 studies from retrospective cohorts to case series of at least four patients per paper according to our methods of research, aiming to describe the clinical presentation, particularly, PHP and non-parathyroid tumours, genetic configuration, and the outcome in HJT amid *CDC73* spectrum [30,31,32,33,34,35,36,37,38,39,40,41,42,43,44,45,46]. In accordance with prior data [16], most patients that were confirmed with the syndrome were usually young with mean age at the onset of disease between 20.8 and 39.5 years, while the largest study found that 71% of patients had the diagnosis before the age of 30 [31]. The early age at presentation strongly indicated a familial form of PHP [49]. Males and females seemed equally affected, unlike sporadic PHP whereas females are diagnosed 3–4 times more often than males [50,51,52]. This is probably due to the germline pathogenic variants of the *CDC73* gene being located on chromosome 1 with autosomal dominant inheritance, irrespective of sex distribution [53]. 

PHP represented the central manifestation in HJT and occurred as the first clinical element in up to 85% of HJT patients [31]. The most frequent cause of PHP was a parathyroid adenoma [30,31,32,33,34,35,36,37,38,39,40,41,42,43,44,45,46]; of note, most genetic PHP as seen in MEN are typically a matter of poly-glandular disease, not a single gland involvement as found in HJT [54,55]. The studies published over the past 10 years (according to our methods) that provided the biochemistry panel found a mean serum calcium level above the level of 12 mg/dL in PHP. Increased serum calcium levels are a common finding in HJT; however, hypercalcemia alone cannot differentiate between HJT and MEN syndromes, as generally known [56,57]. Serum PTH values were elevated in HJT as well, with an average value of at least 236.6 pg/mL [46].

### 3.2. From the CDC73 Gene to Negative Parafibromin Status and Parathyroid Carcinoma

Parathyroid malignancy represents a rare neoplasia, being the cause of <1% of all PHP cases [58]. In HJT, the incidence is much higher, around 15% [59]. In the past 10 years, most large HJT studies published on PubMed reported its diagnosis between the age of 15 [35] and 37.5 years [46]. In some families up to 85% of the carriers suffered from a parathyroid carcinoma [41], indicating that certain pathogenic variants may associate a higher risk. The increased incidence of this neoplasia in HJT is probably related to *CDC73* gene; notably, some of its pathogenic variants have been reported in up to 75% of the sporadic parathyroid malignant tumours, as well (outside HJT) [60,61]. Recognizing parathyroid carcinoma is difficult, still representing a challenge amid modern era. High calcium levels, especially over 14 mg/dL and markedly increased PTH might serve as additional clues [62,63]. Large tumours and ultrasound signs of malignancy such as poorly defined borders, suspicious vascularisation, infiltration, and even the presence of metastasis at first diagnosis may bring information concerning the malignancy status before the actual pathological exam following parathyroidectomy [9,64,65]. Patients with suspected parathyroid carcinomas should undergo surgery, with en-bloc resection and clear margins. Even so, the rate of recurrence may reach 50% [66]. Promoting awareness that this entity is tightly associated with HJT is crucial, considering its incidence is 10 times higher compared with the non-HJT population.

An important issue in HJT is represented by the parafibromin profile in associated tumours. This syndrome is caused by inactivating pathogenic variants of the *CDC73* gene located on chromosome 1q21-q31, leading to an altered expression of parafibromin, a tumour suppressor protein [67]. In sporadic cases, lack of parafibromin stain is mainly found in parathyroid carcinomas [68], while positive staining is highly predictive for a lack of malignancy in parathyroid tumours (prior defined as “adenoma”) [69]. However, in HJT, both parathyroid adenomas and carcinomas might display a deficient immunoreactivity with regard to parafibromin [37,38,70,71]. The absence of nuclear parafibromin staining should therefore raise a high suspicion of a parathyroid carcinoma in HJT [72]. On the other hand, genetic testing for germline *CDC73* variants should be considered in all parafibromin negative parathyroid tumours [9]. Moreover, loss of parafibromin staining was also found in sporadic ossifying fibromas, pointing towards an underlying pathogenic mechanism involving *CDC73*, not only in parathyroid, but also in jaw tumours [73], or in various kidney tumours, pancreatic carcinomas, as well as breast, lung, gastric, and colorectal neoplasia [74,75]. A direct connection between *CDC73* leading to parafibromin deficits and the development of uterine tumours failed to be established yet; however, ovarian carcinoma-associated pathogenesis was linked to parafibromin downregulation, too [76]. Currently, the prevalence of ovarian carcinoma in HJT remains debatable among the other complex genetic, epigenetic and molecular traits in this severe gynaecologic malignancy [77]. 

Overall, parafibromin awareness amid parathyroid tumours in suspected or confirmed cases of HJT represents a very useful indicator with a different significance from sporadic cases. Notably, many of the older studies with respect to parathyroid carcinomas, especially in HJT do not report parafibromin status which represents a major point of interest from now on, despite the current level of statistical evidence in HJT which can only be described as less generous. We anticipate an expansion of parafibromin staining as a mandatory tool in the assessment of HJT-associated tumours (confirmed or suspected to be related to *CDC73* spectrum); its exact clinical relevance as a contributor to the tumour growth and relapse/recurrence or as every day prognostic marker is yet to be defined. 

### 3.3. Non-Parathyroid Syndromic Features in HJT

Other frequent manifestations in HJT are ossifying fibromas of the jaw which are benign tumours developed from the periodontal ligament, being described in 25–50% of HJT subjects [78]. Similarly, we identified a prevalence ranging from 5.4% to 50%, while the largest study found a prevalence of 15.4% [31,34,35,46]. A predisposition for the mandible rather than maxillary bone has previously been observed [78], but tumours may appear both in the mandible and the maxilla [30,31]. The pathogenic mechanisms seem to involve *CDC73* rather than direct PHP complications [30,79]. Their management is mostly surgical [80]. Currently, apart from an increased multidisciplinary disease burden in HJT patients developing jaw tumours and a potential clue for searching serum calcium status in apparently (otherwise) asymptomatic patients who are firstly admitted for a mandible/maxillary neoplasia, the presence of jaw masses does not seem to be correlated with a more severe phenotype regarding HPH (neither a higher risk of a parathyroid carcinoma).

HJT associates a wide variety of renal lesions. Most frequent findings are kidney cysts, with a prevalence of up to 75% [43]. Renal tumours (including cancers) were found in almost 19% of the subjects [46]. A particular manifestation included mixed epithelial and stromal tumours of the kidney [39]. Due to the risk of malignant transformation, surgery is advised in such cases [81]. The diagnosis of kidney tumours should be taken into consideration in patients with HJT, as they may add to the overall burden of the syndrome and participate in the development of kidney failure alongside long-term, uncontrolled (or unrecognized) PHP [82]. However, the specific impact on the renal function is yet to be explored in HJT. Ectopic pelvic kidney has been reported in isolated cases, as well [83].

The risk of uterine lesions seemed increased in HJT, especially with concern to leiomyomas, adenofibromas, and adenomyosis. In addition, malignant uterine lesions such as sarcomas have also been previously reported [84,85]. The underlying pathogenic mechanisms and the involvement of *CDC73* pathogenic variants and parafibromin expression are yet to be explored. Apart from its direct role, parafibromin is linked to other protein interactions and regulates genes connected to cell growth and apoptosis. For example, it is part of PAF1, and by binding to high mobility group AT-hook 1 (Hmga1) and 2 (Hmga2) genes, is involved in chromatin remodelling [86]. This pathway could be further explored. However, there are studies that failed to detect parafibromin expression in normal uterine tissue [87]. On the other hand, in vivo animal models found that heterozygote *CDC73* variant was associated with uterine neoplasms in mice [88]. The cause of uterine lesions, as well as the pathogenic pathways in HJT, needs to be elucidated by further research. 

We may conclude, for practical purposes, that the penetrance of one pathogenic variant in one family does not imply that the entire phenotype involving parathyroid and non-parathyroid tumours should be identically registered in all the other family members across the lifespan, but awareness remains the key factor. 

### 3.4. A Matter of Differential Diagnosis in HJT-PHP

HJT should be differentiated from other forms of familial PHP, as mentioned [89]. A retrospective analysis by Figueiredo et al. [30] on 48 patients found that HJT-PHP was the most prevalent form of familial PHP with a prevalence of 41.7% (N = 20). When compared with other familial types, subjects with HJT had higher PTH levels than individuals with MEN2A, and more elevated calcium values than subjects suffering from MEN1 or MEN2A [30]. A retrospective study by Kong et al. [45] identified 12 kindred with familial PHP. Family out of the twelve had HJT caused by a heterozygous deletion (c.307 + ?513-?del, exons 4, 5, 6) of the *CDC73* gene. Two female members of this family were diagnosed with PHP due to parathyroid adenomas, and parathyroid hyperplasia, respectively (according to the original terms). One of these patients was confirmed with PHP by the age of 15, while the other, at 21 and they were both referred to surgery. Parafibromin staining was investigated in one of them and was absent. Both patients also associated uterine fibroids. Only one woman developed an ossifying fibroma of the jaw [45].

Another important aspect is the differential diagnosis between HJT-PHP and familial isolated PHP with *CDC73* mutations. While both are familial syndromes, familial isolated PHP only associates PHP without other tumours, and some authors considered it as being an incomplete form of HJT [90,91,92]. The risk of developing other tumours should be kept in mind for these patients, too. Currently, the heterogeneous expression of parafibromin status, the wide spectrum of *CDC73* mutations, as well as the variety of clinical presentations make it hard to predict the exact phenotype based on genotype. The central role of PHP is, however, clear, and the increased risk of parathyroid carcinoma under these specific circumstances was confirmed by most studies. 

### 3.5. CDC73: A Strong Player in the Game of HJT

Numerous variants of the *CDC73* gene have been associated with HJT, and novel ones are being identified; however, as already mentioned, the genotype-phenotype correlations remain an open matter, with various unknown and hidden issues, as proven over the years [93,94,95,96,97,98]. The highest odds of having a *CDC73* pathogenic variant were found in subjects younger than 41.5 years [99]. One argument for the heterogeneity of this gene expression is that the same genetic variant resulted in different phenotypes in distinct individuals belonging to the same kindred [31]. Sometimes, *CDC73* pathogenic variants can be associated with others, such as mitochondrial mutations, leading to a particular phenotype, as was the previously cited case in which the co-presence of a germline *CDC73* deletion and mitochondrial somatic pathogenic variants led to the development of oncocytic parathyroid adenoma and carcinoma [33]. Another area for future research is represented by the interactions between ubiquitin-specific protease 37 (USP37) and the *CDC73* gene. In HJT, USP37 was found to be responsible for the stability of *CDC73* [91,100]. The clinical implications are yet to be explored.

### 3.6. CDC73-Related HJT: Lower-Evidence Synthesis

Collaterally, we introduce another sample-based analysis with a less strong statistical support, meaning the case reports (≤three individuals per article) that were identified by using the same methodology in terms of PubMed publications during the last decade which were found according to the already mentioned key research terms. Given the heterogeneity of the clinical and non-clinical features included in HJT diagnosis, we selected case reports with confirmed pathogenic variants of *CDC73* in association with a personal or family history of tumours underlying HJT (we excluded cases without genetic testing or confirmation and non-germline *CDC73* pathogenic variants). 

Thus, we additionally found twenty articles [101,102,103,104,105,106,107,108,109,110,111,112,113,114,115,116,117,118,119,120], a total of 21 individuals with HJT. The age at diagnosis varied between 19 and 60 years, with a mean of 31.9 ± 13 years. 71.43% of them (N = 15) were females [101,102,103,104,105,106,107,108,109,110,111,112,113,114,115,116,117,118,119,120]. Of note, the female predominance was not confirmed by the above mentioned studies on HJT [30,31,32,33,34,35,36,37,38,39,40,41,42,43,44,45,46] that, however, embrace a higher level of statistical evidence. The genetic configuration in these subjects was associated with an interesting *CDC73* spectrum in terms of deletions, such as heterozygous deletion of exons 8–9 (“de novo” variant), heterozygous (large) deletion of exons 1–17, 4.1-Mb deletion on chromosome 1q31.2-31.3, heterozygous c.-16:8del, c.(136_144)del5 or c.18_48del31; insertions, for example, 99-bp insertion in exon 2, p.Asn65_Phe531delinsIleLysTyr, respectively, nonsense variant p.Arg9Stop (R9X) [101,102,103,104,105,106,107,108,109,110,111,112,113,114,115,116,117,118,119,120]. Moreover, a secondary genetic anomaly (c.179 T.A somatic mutation on the other allele) was found at the level of the jaw tumour in a 28-year-old woman [118].

The most frequent clinical manifestation amid HJT was, as expected, PHP, which was reported in 90.47% (N = 19/21) subjects, representing the central clinical element in symptomatic cases. In 52.63% of these persons (N = 10, accounting 47.62% of all HJT patients), the histological profile was a parathyroid adenoma, including one patient (1/19) suffering from two ectopic adenomas [105], and another (1/19) with double orthotopic adenomas [118]. Parathyroid carcinomas occurred in 21% (N = 4) of all PHP cases [110,111,117,119], again, confirming a higher prevalence of this histological type among HJT, which generally concerns less than 1% of all subjects diagnosed with PHP in general population, as specified above [121,122,123]. Moreover, the rate of parathyroid malignancy among the individuals with PHP is similar with the study-based analysis [30,31,32,33,34,35,36,37,38,39,40,41,42,43,44,45,46]. Atypical adenomas (currently, named “atypical tumours”) [9,124,125] were confirmed in 10.5% (N = 2) of PHP cases, representing 9.5% of all patients. Also, in one subject the underlying cause was a parathyroid hyperplasia (currently named “multi-glandular disease”) [9,124,125] in three of the parathyroid glands, along with a cystic nodule [108]. The largest parathyroid tumour was of 5 cm [116]. The typical complications of long standing hypercalcemia and increased PTH were reported:, for instance, premenopausal osteoporosis in young females [101,109] and a middle- aged male with a low bone mineral density [117]. Interestingly, one report of osteitis fibrosa cystica was proved among these 19 cases of PHP, raising the issue of a higher prevalence in HJT-PHP, which is still an open issue [119]. Kidney stones and nephrocalcinosis were not part of the *CDC73*-related renal panel, but are a direct consequence of PHP-associated hypercalcemia [105,115]. 

The second most frequent clinical feature was a jaw tumour (N = 12, 57.14%), that associated different histological reports: ossifying fibromas (N = 9), fibrous dysplasia (N = 1), and a giant cell granuloma (N = 1). The mandible/maxillary ratio with concern to the location of these jaw lesions was of 5 to 4. The largest tumour reached 7 cm in a 19-year-old woman [116]. Uterine lesions were reported in 33.33% of the HJT females (N = 5) and included leiomyomas (N = 2), endometrial polyps (N = 2), and an endocervical glandular polyp (N = 1). Kidney manifestations were found in 28.57% (N = 6) of the individuals, being represented by cysts (N = 3), a renal cell carcinoma (N = 1), nephroblastoma (N = 1), and mixed epithelial and stromal tumours (N = 1) [101,102,103,104,105,106,107,108,109,110,111,112,113,114,115,116,117,118,119,120]. One report included an incidentally detected small thyroid lesion underlying a differentiated type of carcinoma [111]. Papillary neoplasia represents the most frequent histological subtype of thyroid malignancy, noting the increasing incidence of thyroid incidentalomas in modern medical era [126,127]. Nowadays, no direct link to *CDC73* or parafibromin profile is sustained. Of note, one mentioned observational study also identified a case of papillary micro-carcinoma of the thyroid [34].

Two out of the 15 female patients with HJT presented ovarian lesions, namely bilateral ovarian cysts [107], and an ovarian granulosa cell tumour [112]. The connection with *CDC73* spectrum represents a further topic to explore. So far, the level of statistical evidence remains low, and the pathogenic loops are unknown; however, with regard to the gynaecological cancers (including breast malignancy), *CDC73*-mediated apoptosis via HECT domain E3 of ubiquitin ligase might play a critical role [128]. 

Data regarding parafibromin status were scarce: weak parafibromin staining in carcinoma nuclei belonging to a parathyroid carcinoma were detected on a 41-year-old man [110], as well as decreased parafibromin staining in the jaw tumour of a young female was reported [118]. Routine parafibromin stain in maxillary and mandible masses amid HJT is currently incompletely implemented in daily practice, having yet an indeterminate significance. Parafibromin is part of the new wave concerning immunohistochemistry-based markers of different endocrine and non-endocrine tumours, particularly in hereditary syndromes [129]. Moreover, further exploration of molecular profiling is mandatory, in addition to understanding the role of epigenetic mechanisms (such as histones modification and DNA methylation) interplaying with *CDC73*-related tumours (Table 4) [130,131,132].

### 3.7. From Genetics to a Practical Approach in HJT

The management of *CDC73*-related HJT is complex, and it requires a multidisciplinary team, as it affects not only the parathyroid glands, but also the jaw, the kidneys, and the uterus. The highest danger is posed by the elevated prevalence of parathyroid carcinoma [9,116]. PHP was treated surgically in all studies according to our 10-year retrospective, regardless of the tumour type. Considering that most often there was a single-gland involvement, selective parathyroidectomy was preferred [30,31,32,33,34,35,36,37,38,39,40,41,42,43,44,45,46]. Whether the latest past 4-year period affected these specific endocrine areas and the associated panel of endocrine surgery amid the COVID-19 pandemic, as generally seen in different aspects of medicine and surgery, is difficult to appreciate in HJT domain [133,134,135,136].

The reliability of preoperative localisation in HJT-PHP is variable, and the data regarding the best surgical approach remains rather limited so far. Unilateral exploration based on preoperative findings, as well as bilateral neck exploration have been suggested [35,46]. The risk of recurrence is high, especially in patients with parathyroid carcinoma, due to inconclusive genotype-phenotype correlations and lack of specific prognostic markers thus the surgical management should be individualised [137,138]. Most frequently, HJT-PHP was cured by the surgical management. There were cases, however, where surgery did not provide disease-free evolution, with a recurrence rate of 23.5% in parathyroid adenomas and 66% in parathyroid carcinomas [35]. Larger studies are needed for more accurate estimates. The post-operatory complications were similar to the usual ones [102], including recurrent nerve palsy and transient or permanent hyperparathyroidism. A higher risk of permanent hyperparathyroidism in patients with subtotal parathyroidectomy was found in one study [46].

The mentioned studies [30,31,32,33,34,35,36,37,38,39,40,41,42,43,44,45,46] were not generous with concern to the metabolic and bone features, as generally expected in children and adults with PHP, as well as the role of vitamin D status amid HJT-PHP and potential epigenetic role of its deficiency, including *CDC73* interplay [139,140]. Osteoporosis associated with fragility fractures has been reported in patients younger than 21 years, as mentioned [101]. Considering the early PHP onset in most HJT individuals, and the fact that HJT may manifest in children, we should be mindful of the impact of PHP and hypercalcemia on skeleton status (including peak bone mass) [141,142,143]. Yet, no specific controlled study addressed the issue of secondary osteoporosis in HJT-PHP, which currently is managed according to general recommendations for PHP-related osteoporosis and hereditary PHP [142,143,144,145,146]. Of note, specific paediatric studies in the matter of HJT-PHP are deficient. 

As far as the limits of the current sample-based synthesis are concerned, we mention the ne-systematic review that was, however, meant to cover the most recent data with respect to different domains connected to HJT field and its clinical, genetic, and molecular issues which, for the moment, embrace various levels of statistical evidence. As pointed out, larger cohorts are needed to pinpoint the exact influence of *CDC73* and parafibromin profile on clinical picture, specific algorithms of detection and lifelong follow-up, including awareness of the non-endocrine conditions located at jaw, kidney, and uterus, as well as immunohistochemistry and molecular indices to serve as prognostic factors. Noting these 17 studies [30,31,32,33,34,35,36,37,38,39,40,41,42,43,44,45,46] and the collateral analysis of another 20 case reports [101,102,103,104,105,106,107,108,109,110,111,112,113,114,115,116,117,118,119,120], we summarise below the main findings in HJT (Figure 1).

## 4. Conclusions

HJT represents an important chapter of the parafibromin deficient parathyroid neoplasms that have been recently recognised as a distinct category and their importance is under evaluation through different clinical, endocrine, pathogenic, and surgical aspects. When faced with familial cases of PHP, especially with single gland involvement, negative nuclear parafibromin staining is a useful tool guiding further *CDC73* genetic testing. However, it ought to be taken into consideration that positive parafibromin staining does not exclude HJT. If clinical suspicion is high, *CDC73* testing is still needed. HJT should also come to mind during the management of patients with ossifying fibromas of the jaw, especially in young patients, as it can sometimes be the first manifestation of this syndrome. Considering that the patients suffering from HJT are often young, the high risk of developing parathyroid carcinoma should not be overlooked. Prompt diagnosis and adequate genetic counselling, as well as optimal management might improve the overall disease burden. The wide phenotypical spectrum of *CDC73*-associated disorders requires further genotype-phenotype correlation studies.

## Figures and Tables

**Figure 1 ijms-25-02301-f001:**
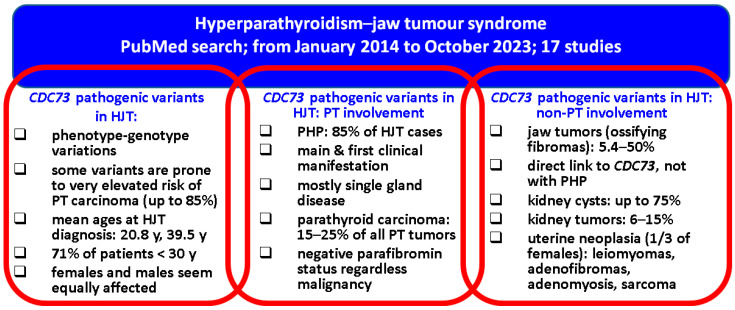
Qualitative analysis in HJT: from the *CDC73* gene to parathyroid and non-parathyroid findings [30,31,32,33,34,35,36,37,38,39,40,41,42,43,44,45,46,101,102,103,104,105,106,107,108,109,110,111,112,113,114,115,116,117,118,119,120].

**Table 1 ijms-25-02301-t001:** Original human studies (N ≥ four subjects/article) addressing HJT and *CDC73* pathogenic variants-associated clinical considerations amid past decade according to our mentioned methods (starting with the most recent publication date) [30,31,32,33,34,35,36,37,38,39,40,41,42,43,44,45,46].

Author/Reference Number/Year	Study DesignNumber of PatientsStudied Population	Genetic Germline Variant of *CDC73* Gene	Phenotype
Figueiredo2023[30]	**Retrospective study**N = 48**HJT**: N = 20Age at onset of symptoms: mean = 33.3 ± 13.1 yAge at onset of **PHP**: mean = 35.1 ± 13.6 ySex ratio: 1:1 (10 males and 10 females)	c.356del, p.(Gln119Argfs*14)55% (N = 11)	Prevalence of **PHP** as first manifestation: 85% (N = 17/20) of patients with HJT**JT** as first manifestation in 10% (N = 2)**PC** in 23.5% (N = 4/17)
c.766_767del, p.(Val256Lysfs*10)30% (N = 6)
c.306G > A, p.(N = 1)
c.520_523del, p.(Ser174Lysfs*27)(N = 1)
Whole gene deletion (N = 1)
Tora2023[31]	**Retrospective study**N = 68**PHP**: N = 55Age at diagnosis: 10–30 y (in 71%)53% (N = 36) males and 47% (N = 32) females	Biallelic loss: germline *CDC73* variant p.L95P and pathogenic variant in the trans allele, p.M1V (c.1A > G)	Wilms tumour	**PHP** as first manifestation: 85% (adenoma in 65% and carcinoma in 31%)Diagnosis of **PHP** based on:  Active surveillance of calcium levels due to family history (33%)  Routine blood work (22%)  Symptoms (35%)**JT** as first manifestation in 5% (N = 3) of symptomatic patients → 57% (N = 4/7) of **JT** were surgically removed**PC** in 31% (N = 17/55)
c.3G > T (p.M1I)	Mixed epithelial stromal tumours
c.164 A > C (p.Y55S)	Kidney tumours
Yang2022[32]	**Case series**N = six carriers of the same family**PHP**: N = 4Age at diagnosis in symptomatic patients: 31, 33, 36, and 58 y50% (N = 3) males and 50% (N = 3) females	130 kb deletion spanning exon 1–6 of *CDC73* and the upstream UCHL5, TROVE2, and GLRX2i	**PHP**: 67% (N = 4/6)**PC**: one case (representing 16.7% of carriers)
De Luise2021[33]	**Case series**N = four members of the same familyAges at diagnosis: 37, 42, 47, and 62 y	Deletion of the first 10 exons of *CDC73*and pathogenic germline variant m.2356A > G of the mitochondrial DNA, somatic mitochondrial pathogenic variants (m.14973G > A, m.5147G > A, m.3380G > A, m.14387A > G, m.10371G > A)	**PC** (oncocytic type): N = 2**PA** (oncocytic type): N = 2
Le Collen2021[34]	**Observational study**N1 = 47 members of the same familyN2 = 13 *CDC73* variant carriersAge at onset of symptoms: median = 37 ± 13.6 yAge at onset of **PHP**: 13–54 y (median = 39 ± 14.5 y; mean = 38 y)**PHP**: 71.4% (N = 5) males, and 28.6% (N = 2) females**JT**: 50% (N = 1) males and 50% (N = 1) females	c.(237 + 1_238-1)_(307 + 1_308-1)del;p.	**PHP**: N = 7 **JT**: N = 2, sole manifestation in 7.52% (N = 1)Healthy carriers (N = 4)Renal complications of hypercalcemia in 57.1% (N = 4):  Urinary lithiasis  Nephrocalcinosis  Chronic kidney failure
Iacobone2020[35]	**Retrospective study**N = 37**PHP**: N = 22Age at PHP onset: 11–71 y (median = 38 y)40% (N = 8) males and 60% (N = 14) females	Frameshiftc.433_442delinsAGA (N = 8/14)	**PT**: N = 7**JT**: N = 1Uterine involvement: N = 5/5	Prevalence of **PHP** as first manifestation: 59.5% (N = 22/37) of patients with **HJT** **JT**: 5.4% (N = 2)**PC**: 15% (N = 3/20)
Missense c.188T>C N = 7/19	**PT**: N = 6Uterine involvement: N = 4/8
c.136_144 del5 (N = 7/9)	**PT**: N = 6**JT**: N = 1Uterine involvement: N = 3/5
Frameshift c.276delA, p.Asp93Ilefs*16 (N = 6/8)	**PT**: N = 2Uterine involvement: N = 2
5′UTR: c.-2insG (g.5182insG) (N = 3/3)	**PT**: N = 1
Li 2020[36]	**Retrospective study** of two cohorts: one original cohort (N = 68) and a validation cohort from reported cases (N = 351)	High-impact (gross indels, splicing, frameshift, and nonsense) germline variants and variants affecting the C-terminal domain of the parafibromin protein are associated with:  Higher risk of **PC**  Higher risk of **JT**  Larger tumour size  Higher levels of PTH	**PC**: 20.6% (N = 14)**PA**: 61.8% (N = 42)17.6% (N = 12) did not have any parathyroid disease
Gill2019[37]	**Retrospective study**N = 40 subjects with 43 parafibromin-negative tumours(16 of them had *CDC73* variants)	c.685_688delAGAG, p.(Arg229Tyr)fs*27 (N = 1)IVS7 + 2T > G (N = 1)c.271C > T, p.(Arg91*) (N = 1)Whole-gene deletion (N = 1)	**PC** and loss of parafibromin staining: N = 4
c.1247delG, p. (Gly416Ala)fs*12 (N = 1)c.226C > T, p. (Arg27*) (N = 1)c.157 G > T, p. (Glu53*) (N = 3)c.415 C > T, p. (Arg139*) (N = 1)c.226 C > T, p. (Arg76*) (N = 3)IVS2 + 1G > C (N = 1)c.271C > T, p. (Arg91*) (N = 1)Whole-gene deletion (N = 1)	**PA** and loss of parafibromin staining: N = 12
Grigorie2019[38]	**Case series**N = 3/6 of the same familyAges at diagnosis: 24, 35, and 46 y 67% (N = 2) males and 33% (N = 1) females	c.128-IVS1 +1 delG with polymorphism rs41302543	**PC**Jaw ossifying fibromaUterine fibroid	**PHP**: 50% (N = 3) of carriers: **PC** (N = 2) and double **PA** (N = 1)
c.128-IVS1 +1 delG with polymorphism rs80356645	**PC**
c.128-IVS1 +1 delG with polymorphism rs4466634	Double **PA**
Vocke2019[39]	**Case series** (one family member screening)N = three family members confirmed with germline *CDC73* mutation Age at diagnosis: 45, 47, and 70 y 33% (N = 1) male and 67% (N = 2) females	c.3G > T	Mixed epithelial and stromal tumours of the kidney in all carriers (N = 3)Uterine fibroids in all female carriers (N = 2)
Guarnieri2017[40]	**Case series**N = 3/4 members of the same familyAge at diagnosis in proband: 48 y33% (N = 1) male and 67% (N = 2) females	c.-4_-11insG variant within the 5′UTR associated with a microdeletion of 0.25 Mb in band 1q31.2	**PC**: N = 1**PA**: N = 2Uterine leiomyoma: N = 1
Mamedova2017[41]	**Observational study****PHP**: N = 65**HJT**: N = 6Age at diagnosis: 13–24 y16.7% (N = 1) males and 83.3% (N = 5) females	c.271C > T, p.Arg91	**PC**	**PC**: 67% (N = 4/6)Atypical **PA**: 17% (N = 1)
c.496C > T, p.Gln166	**PC**
c.685A > T, p.Arg229	**PC**
Gross deletion of the whole *CDC73* gene	**PC**
Deletion of 1–10 exons	Atypical **PA**
c.787C > T, p.Arg263Cys	Solitary parathyroid hyperplasia
van der Tuin2017[42]	**Retrospective study****PHP** referred for genetic testing of *CDC73*: N = 89 → **HJT**: N = 18 → pathogenic germline *CDC73* in 17% (N = 3)Positive relatives: N = 27	c.687_688dellAG, p.(Arg229Serfs*37) in 65% (N = 24) of relatives of the index case	**PA**: N = 14**JT**: N = 2Renal cysts: N = 5Pancreatic ductal adenocarcinoma: N = 2Hürthle cell adenoma of the thyroid: N = 1Mixed germ cell testicular tumour: N = 1
c.3_15dup, p.(Ser6Glyfs*5) in all relatives (N = 3)	**PA**: N = 1**JT**: N = 2Congenital urinary tract abnormalityWilms tumour
c.760C.T, p.(Gln254*) in 50% of relatives (N = 4)	**PA**: N = 1**JT**: N = 1
Khadilkar2015[43]	**Case series**N1 = seven patients with **PHP** caused by *CDC73* variantsMean age at presentation for index cases with **PHP**: 27.25 ± 9.8 yN2 = four patients with **HJT** Ages at diagnosis in **HJT**: 20, 40, 30, and 54 y50% males and 50% females	c.14_17dupTTAG	**PC****JT**Kidney cysts
c.40C > T at codon 14	Cystic **PA** **JT** Renal cystsEndometrial polyp
c.415C > T at codon 139	**PA**Renal cystsUterine fibroids
Chiofalo2014[44]	**Case series**N = four members of the same familyCarriers: N = 2/4Affected members N = two (females)Age at diagnosis (proband): 28 y	c.1379delT	Parathyroid lesions 50% (N = 2)Ossifying fibroma of the jaw 25% (N = 1)
Kong2014[45]	**Retrospective study**Familial **PHP**: N = 22**HJT**: N = 2Proband diagnosis at age of 15 y	c.307 + ?_513-?del exons 4, 5, 6	**PA**: N = 1Parathyroid hyperplasia: N = 1Ossifying fibroma of the jaw: N = 1Uterine fibroids: N = 2
Mehta2014[46]	**Retrospective study**N = 16Age at diagnosis: 18–49 y62.5% (N = 10) males and 37.5% (N = 6) females	Whole-gene deletion(N = 7)	**PA**: N = 3**PC**: N = 4	Symptomatic **PHP** (100%):  Neurological manifestations (headaches, memory loss, difficulty concentrating): 43.8%  Fatigue: 37.5%  Gastrointestinal symptoms (abdominal pain, constipation, gastroesophageal reflux): 31.3%  Polyuria and/or nephrolithiasis: 18.8%  Bone pain: 18.8%  Fractures: 12.5%**JT**: 12.5% (N = 2)**PC**: 37.5% (N = 6)
c.687_688dupAG(N = 5 pertaining to 3 families)	**PA**: N = 3**PA**: N = 2
p.Tyr55Ser(N = 2)	**PA**
c.664C > T, p.Arg222X(N = 1)	**PA**
c.226C > T, p.Arg76X(N = 1)	**PA**

**Abbreviations**: HJT = hyperparathyroidism–jaw tumour syndrome; JT = jaw tumour; PC = parathyroid carcinoma; PA = parathyroid adenoma; PT = parathyroid tumour; PHP = primary hyperparathyroidism; PTH = parathyroid hormone; N = number of patients; y = year; of note, the description of the studied population (including provided ages and sex distribution) as well as the histological report of the parathyroid tumour has been determined according to the original data (if available).

**Table 2 ijms-25-02301-t002:** Lab findings (mean serum total calcium and average PTH) that confirm primary hyperparathyroidism in HJT according to our methods of literature research (if available; according to the original cited data); the display starts with the lowest mean calcaemic values to the highest [32,34,35,41,43,46].

Reference Number	Mean Calcium Levels	Mean PTH Levels
[32]	10.62 mg/dL (2.65 mmol/L)	411.25 pg/mL
[46]	12.20 mg/dL	236.6 pg/mL
[34]	12.54 ± 2.8 mg/dL (3.13 ± 0.7 mmol/L)	115 ± 406 pg/mL
[43]	12.75 mg/dL	838.75 pg/mL
[35]	12.90 mg/dL (3.24 mmol/L)	2.29-fold increase (*)
[41]	15.00 mg/dL (3.74 mmol/L)	1324.65 pg/mL

* According to the original study.

**Table 3 ijms-25-02301-t003:** Overview of non-parathyroid, non-jaw tumours in HJT according to our methods [31,34,35,39,42,43,46] (N = number of patients per study).

Tumour Type	Studied Population	Reference Number
Uterine tumours	38%	[31]
	33% (N = 2/6)	[46]
	60.8% (N = 14)	[35]
	N = 1	[34]
	100% (N = 2)	[43]
	100% (N = 2)	[39]
Kidney tumours	6%	[31]
	18.8% (N = 3/16)	[46]
	2.7% (N = 1)	[35]
Kidney cysts	15.4% (N = 2)	[34]
	75% (N = 3)	[43]
	N = 5	[42]
Mixed epithelial and stromal tumours of the kidney	N = 3	[39]
Papillary thyroid micro-carcinoma	N = 1	[34]
Prostate cancer	N = 1	[34]
Pancreatic ductal adenocarcinomas	N = 2	[42]
Hürthle cell adenoma of the thyroid	N = 1	[42]
Mixed germ cell testicular tumour	N = 1	[42]

**Table 4 ijms-25-02301-t004:** Overview of HJT case reports (germline *CDC73* pathogenic variants) according to our methods; starting with the most recent publication date (from 2023 to 2014) [101,102,103,104,105,106,107,108,109,110,111,112,113,114,115,116,117,118,119,120].

First Author/Reference Number/Year	Studied Patients	Germline Variant of *CDC73* Gene	Parathyroid Tumour’s Features	Jaw Tumour	Kidney Findings *	Uterine Findings	Other Observations
Danda2023[101]	21-year-old female	99-bp insertion in exon 2, p.Asn65_Phe531delinsIleLysTyr	**PA** (US: 3.8 cm × 1.4 cm × 2.2 cm)		CystsEctopic pelvic kidney		Osteoporosis
Kang2023[102]	60-year-old male	Pathogenic variant c.376C > T, p.Arg126	Atypical **PA**		Bilateral renal cysts	NA	Recurrent hypercalcemia 10 years after parathyroid surgery
Garrigues2022[103]	28-year-old female	Heterozygous pathogenic variant c.687_688dup, p.Val230Glufs*28	**PA** (0.9 cm)				Proband’s mother: **PHP** and uterine anomalies
Yang2022[104]	32-year-old female	Heterozygous c.1A > G, p.Met1Val		Ossifying fibromas of the mandible	Mixed epithelial and stromal tumours (4.1 cm × 5.2 cm and 4.9 cm × 4.6 cm)	 Leiomyomas  Endometrial hyperplasia	
56-year-old male	Heterozygous c.1A > G, p.Met1Val		Recurrent ossifying fibromas of the mandible and bilateral maxillary		NA	
Barnett2021[105]	36-year-old female	Heterozygous deletion of exons 8–9 (“de novo” variant)	Large ectopic **PA** + a second ectopic **PA**		Multiple cysts	Multiple leiomyomas	Nephrocalcinosis
Gupta2021[106]	Male in his late teens	Pathogenic truncating variant c.25C > T, p.Arg9Ter	**PHP** referred for surgery		Renal cell carcinoma	NA	
Weaver2021[107]	23-year-old female	Unnamed pathogenic variant	**PA**				Bilateral ovarian cysts
Arfi 2020[108]	32-year-old female	Heterozygous deletion	Three hyperplastic parathyroid glands and a cystic parathyroid nodule			 Endocervical glandular polyp  Polypoid adenomyomatous endometrium	Psoriasis
Wijewickrama2020[109]	22-year-old female	Heterozygous large deletion of exons 1–17	**PA** (CT: 2.5 cm × 2.9 cm)	Maxillary tumour (fibrous dysplasia)	Nephroblastoma		Osteoporosis
Ciuffi2019[110]	41-year-old male	Heterozygous c.191-192 delT	**PC**: US: 3.5 cm × 2 cm × 1 cm)	Ossifying fibroma		NA	Weak parafibromin staining in carcinoma nuclei
Russo2019[111]	42-year-old female	c.580A > T, p.Arg194	**PC**	Ossifying fibroma of the maxilla (5 cm × 3 cm × 4 cm)			Incidental papillary thyroid carcinoma (of 0.4 cm)
Sirbiladze2019[112]	31-year-old female	Pathogenic variant c.687_688dupAG, p.Val230Glufs*28	**PA**				Ovarian granulosa cell tumourPatient’s mother: **PHP**, renal cysts
Koikawa 2018[113]	20-year-old female	Change of CGA (Arg) to TGA (Stop) at codon 126 in exon 5	Parathyroid “mass” (1.5 cm × 1.1 cm × 0.6 cm)			Uterine “mass”	
Rubinstein2017[114]	32-year-old female	4.1-Mb deletion on chromosome 1q31.2-31.3	**PA** (2 cm)	Ossifying fibroma of the mandible (1.7 cm)			
Bellido2016[115]	23-year-old female	Heterozygous c.-16:8del	**PA** (3 cm)	Bilateral ossifying fibromas			Nephrolithiasis
Mathews2016[116]	19-year-old female	Nonsense variant p.Arg9Stop (R9X)	Atypical **PA** (5 cm × 2.3 cm × 1.7 cm)	Ossifying fibroma of the maxilla (7 cm)			
Mele2016[117]	41-year-old male	c.358C > T, p.R120X	**PC** (US: 5.1 cm × 2 cm × 2.2 cm)	Yes			OsteoporosisFour asymptomatic carriers in his family
Masi2014[118]	28-year-old female	c.(136_144)del5	Two **PAs**	Ossifying fibroma (3 cm)		Endometrial hyperplastic polyps	c.179 T.A somatic mutation on the other allele in the jaw tumourDecreased parafibromin staining in the jaw tumour
Parfitt2014[119]	27-year-old male	c.18_48del31	**PC**	Bilateral ossifying fibromas of the mandible (3 cm × 1.4 cm × 2.3 cm and 1.1 cm × 0.5 cm)		NA	Osteitis fibrosa cystica
Reddy2014[120]	25-year-old female	c.18_48del31	**PA** (0.5 cm)	Giant cell granuloma of the mandible			

Abbreviations: cm = centimetre; CT = computed tomography; US = ultrasound; NA = not available; PA = parathyroid adenoma; PHP = primary hyperparathyroidism; of note, pathological report is concordant with the original data; * the renal findings that were found (other than PHP complications); blue = complications of PHP.

## Data Availability

Not applicable.

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
