# Peer review of "Insights into Hyperparathyroidism–Jaw Tumour Syndrome: From Endocrine Acumen to the Spectrum of CDC73 Gene and Parafibromin-Deficient Tumours"

_ijms, 2024, doi:10.3390/ijms25042301_

Round 1
Reviewer 1 Report
Comments and Suggestions for Authors
Dear Editor,
I reviewed the study titled " Insights into hyperparathyroidism-jaw tumour syndrome: from endocrine acumen to the spectrum of CDC73 gene and parafibromin-deficient tumours". Gheorghe et al. have put together a review of the CDC73 genetic test and the challenges and pitfalls of parafibromin staining, aiming to raise more awareness of HJT from the clinical perspective of PHP. They identified 17 original human studies in ten years. They found that HJT was usually diagnosed before the age of 30, and unlike sporadic PHP, men and women seemed to be equally affected. PHP represented the central manifestation of HJT and appeared as the first symptom in up to 85% of HJT cases. The incidence of parathyroid carcinoma was much higher in HJT than in non-HJT patients. In some families, up to 85% of carriers suffer from parathyroid carcinoma, suggesting that some pathogenic variants may carry a higher risk for the CDC73 gene spectrum. Congratulations to the authors. I think it will be useful for clinicians.
Sincerely
Comments on the Quality of English LanguageModerate editing of English language required
Author Response
Response to Reviewer’ Comments – Reviewer number 1
Dear Reviewer,
Thank you very much for your time and your effort to review our manuscript.
We are very grateful for providing your valuable feedback on the article.
Here is our response and related amendment that has been made in the manuscript according to your review (marked in yellow colour).
Point 1
Dear Editor,
I reviewed the study titled "Insights into hyperparathyroidism-jaw tumour syndrome: from endocrine acumen to the spectrum of CDC73 gene and parafibromin-deficient tumours". Gheorghe et al. have put together a review of the CDC73 genetic test and the challenges and pitfalls of parafibromin staining, aiming to raise more awareness of HJT from the clinical perspective of PHP. They identified 17 original human studies in ten years. They found that HJT was usually diagnosed before the age of 30, and unlike sporadic PHP, men and women seemed to be equally affected. PHP represented the central manifestation of HJT and appeared as the first symptom in up to 85% of HJT cases. The incidence of parathyroid carcinoma was much higher in HJT than in non-HJT patients. In some families, up to 85% of carriers suffer from parathyroid carcinoma, suggesting that some pathogenic variants may carry a higher risk for the CDC73 gene spectrum. Congratulations to the authors. I think it will be useful for clinicians.
Sincerely
Thank you so much. We really appreciate it!
Point 2
Moderate editing of English language required.
Thank you. We re-edited the English language. Thank you
Thank you so much.
Reviewer 2 Report
Comments and Suggestions for Authors
Line 122: 'parafibromin’s role, as reflected by the expression “para this, fibromin that”, specifically with regard to the HJT profile' - I do not understand the meaning.
Lines 128-132: It seems to be systematic not narrative review. I suggest to add the 'a systematic review' in the title.
Edit Table 1, please (sympthoms and age of the onset in separate collumn according pathology type, change font size, use only black leters if possible, add number CDC73 positive (+) and negative (-) cases in immunohistochemistry).
Table 3. It is obvious that uterine tumouts belong to femal population - clarify what you mean here.
Figure 1. Correct sizes of marked areas (they are merged) or change figure to table.
What about dental caries and sialolithiasis in HJT?
Comments on the Quality of English LanguageSome sentences are too long.
Line 95: ',however...' => 'however, ...'
Table 1: Add space between number of cases (e.g. 'N=20' => 'N = 20'). Write M: F ratio as 1:1 rather than 1, please (it would look better).
Line 721: 'from 2023 to 2014' => 'from 2014 to 2023'
Author Response
Response to Reviewer’ Comments – Reviewer number 2
Dear Reviewer,
Thank you very much for your time and your effort to review our manuscript.
We are very grateful for providing your valuable feedback on the article.
Here is our response and related amendment that has been made in the manuscript according to your review (marked in yellow colour).
Point 1
Line 122: 'parafibromin’s role, as reflected by the expression “para this, fibromin that”, specifically with regard to the HJT profile' - I do not understand the meaning.
Thank you very much. This means unexpected and divergent implications of the mentioned gene and associated molecular loops. We added to the main text the followings: “para this, fibromin that” (thus showing the complexity and unexpected implications and interferences), specifically with regard to the HJT profile”.
Thank you
Point 2
Lines 128-132: It seems to be systematic not narrative review. I suggest to add the 'a systematic review' in the title.
Thank you. Indeed, this is a narrative review, not a systematic one. Due to the heterogeneity of the spectrum in CDC073-HJT field, we choose to introduce the data as a narrative review since various levels of statistical evidence are identified in the mentioned and cited papers as pointed out across the article. On the other hand, a systematic review pinpoints a specific critical assessment which in the matter of CDC73-HJT is rather limited so far. However, this type of review is a well-recognized, standard, traditional approach which is suitable for topics with less generous publications as seen here. This allowed us to examine and evaluate the scientific panel on a larger and more complex basis. We mentioned at Discussion this particular aspect of this research/paper approach. “As limits of the current sample-based synthesis we mention the ne-systematic review that was, however, meant to cover the most recent data with respect to different domains connected to HJT field and its clinical, genetic, and molecular issues which, for the moment, embrace various levels of statistical evidence.”
Thank you
Point 3
Edit Table 1, please (symptoms and age of the onset in separate column according pathology type, change font size, use only black letters if possible, add number CDC73 positive (+) and negative (-) cases in immunohistochemistry).
Thank you very much. The symptoms are not registered in this table since it is already too complicated. The age remained in the second column according to the original papers we cited. We reduced the Font size, as you requested. We removed the non-black colour according to your recommendation. The suggested immunohistochemistry report is not available for many studies, thus, by partially introducing it, we introduce a potential bias since not all the patients had this exam done (which is rather new and it is not a standard care in many centres). Thank you
Point 4
Table 3. It is obvious that uterine tumours belong to female population - clarify what you mean here.
Thank you very much. We corrected it. Thank you
Point 5
Figure 1. Correct sizes of marked areas (they are merged) or change figure to table.
Thank you very much. The idea of merging them is that HJT embraces both PT and non-PT involvement and the PT group is related to the non-PT group and the CDC73-HJT spectrum. However, PT is the central piece of the entire figure due the clinical identification in most cases. Thank you
Point 6
What about dental caries and sialolithiasis in HJT?
Thank you very much. According to the mentioned methods, we found no specific data with concern to these interesting aspects you mentioned; this seems like a further topic to explore from now on. Thank you
Point 7
Comments on the Quality of English Language
Some sentences are too long.
Thank you very much. We corrected them. Thank you
Point 8
Line 95: ',however...' => 'however, ...'
Thank you very much. We corrected it. Thank you
Point 9
Table 1: Add space between number of cases (e.g. 'N=20' => 'N = 20').
Write M: F ratio as 1:1 rather than 1, please (it would look better).
Thank you very much. We corrected all of them in Table 1 and 3. Thank you
Point 9
Line 721: 'from 2023 to 2014' => 'from 2014 to 2023'
Thank you very much. We respectfully mention that the table-based display starts with the year 2023, so was the literature research. Thank you
Thank you very much.
Reviewer 3 Report
Comments and Suggestions for Authors
The authors provide a clear and concise overview of hyperparathyroidism-jaw tumour syndrome (HJT), covering its clinical presentation, genetic basis, and associated tumours. They focus on data published in the last decade, ensuring the review is up-to-date with the latest findings, and describe the challenges and pitfalls associated with CDC73 genetic testing and parafibromin staining, which are important considerations for clinicians. The manuscript is well-written and informative and raises important points for further discussion. There are still some concerns, as follows.
#1. The manuscript is long. The authors should at least shorten the abstract to 200 words. The second section (Sample) should be also shortened because there are detailed tables.
#2. A graphical scheme for the CDC73 gene with an indication of genetic variants will help readers understand.
#3. There is no “section 3” in the manuscript. “4. Discussion” should be “3. Discussion.”
Comments on the Quality of English LanguageI do not have specific comments.
Author Response
Response to Reviewer’ Comments – Reviewer number 3
Dear Reviewer,
Thank you very much for your time and your effort to review our manuscript.
We are very grateful for providing your valuable feedback on the article.
Here is our response and related amendment that has been made in the manuscript according to your review (marked in yellow colour).
Point 1
The authors provide a clear and concise overview of hyperparathyroidism-jaw tumour syndrome (HJT), covering its clinical presentation, genetic basis, and associated tumours. They focus on data published in the last decade, ensuring the review is up-to-date with the latest findings, and describe the challenges and pitfalls associated with CDC73 genetic testing and parafibromin staining, which are important considerations for clinicians. The manuscript is well-written and informative and raises important points for further discussion.
Thank you so much. We really appreciate it!
Point 2
There are still some concerns, as follows.
Thank you very much. We followed your recommendations. Thank you
Point 3
#1. The manuscript is long. The authors should at least shorten the abstract to 200 words. The second section (Sample) should be also shortened because there are detailed tables.
Thank you very much. We reduced the length of the abstract. We respectfully mention that MDPI rules do not limit the length of the manuscript that is why we intended to cover a large area of data. The mentioned section includes several subsections in order to make it more readable
- Sample-focused analysis
2.1. HJT syndrome: patients’ characteristics at presentation
2.2. HJT-associated PHP
- Lab findings in PHP
- PHP management: from preoperative localization to
parathyroidectomy
- Parathyroid carcinoma
- Parafibromin staining
- HJT-related jaw tumours
- Other tumours in HJT
- Genetic spectrum in HJT
Thank you
Point 4
#2. A graphical scheme for the CDC73 gene with an indication of genetic variants will help readers understand.
Thank you very much. This is an excellent suggestion, indeed; however, there are so many variants that a simple graphical representation is not enough that is why the third column of the table introduces this heterogeneous panel. Thank you
Point 5
#3. There is no “section 3” in the manuscript. “4. Discussion” should be “3. Discussion.”
Thank you very much. We corrected them. Thank you
Point 6
Comments on the Quality of English Language
I do not have specific comments.
Thank you very much.